# Structures and reactivity of peroxy radicals and dimeric products revealed by online tandem mass spectrometry

Sophie Tomaz [1,8], Dongyu Wang[2,8], Nicolás Zabalegui[3,4], Dandan Li[1], Houssni Lamkaddam[2], Franziska Bachmeier[5], Alexander Vogel [5], María Eugenia Monge [3], Sébastien Perrier[1], Urs Baltensperger[2], Christian George [1], Matti Rissanen [6,7], Mikael Ehn [6], Imad El Haddad [2] & Matthieu Riva [1✉]

Organic peroxy radicals ($RO_2$) play a pivotal role in the degradation of hydrocarbons. The autoxidation of atmospheric $RO_2$ radicals produces highly oxygenated organic molecules (HOMs), including low-volatility ROOR dimers formed by bimolecular $RO_2 + RO_2$ reactions. HOMs can initiate and greatly contribute to the formation and growth of atmospheric particles. As a result, HOMs have far-reaching health and climate implications. Nevertheless, the structures and formation mechanism of $RO_2$ radicals and HOMs remain elusive. Here, we present the in-situ characterization of $RO_2$ and dimer structure in the gas-phase, using online tandem mass spectrometry analyses. In this study, we constrain the structures and formation pathway of several HOM-$RO_2$ radicals and dimers produced from monoterpene ozonolysis, a prominent atmospheric oxidation process. In addition to providing insights into atmospheric HOM chemistry, this study debuts online tandem MS analyses as a unique approach for the chemical characterization of reactive compounds, e.g., organic radicals.

[1] Univ Lyon, Université Claude Bernard Lyon 1, CNRS, IRCELYON, 69626 Villeurbanne, France. [2] Laboratory of Atmospheric Chemistry, Paul Scherrer Institute, 5232 Villigen, Switzerland. [3] Centro de Investigaciones en Bionanociencias (CIBION), Consejo Nacional de Investigaciones Científicas y Técnicas (CONICET), Godoy Cruz 2390, C1425FQD Ciudad de Buenos Aires, Argentina. [4] Departamento de Química Inorgánica, Analítica y Química Física, Facultad de Ciencias Exactas y Naturales, Universidad de Buenos Aires, Ciudad Universitaria, C1428EGA Buenos Aires, Argentina. [5] Institute for Atmospheric and Environmental Sciences, Goethe-University Frankfurt, 60438 Frankfurt am Main, Germany. [6] Institute for Atmospheric and Earth System Research, INAR /Physics, Faculty of Science, University of Helsinki, FI–00014 Helsinki, Finland. [7] Aerosol Physics Laboratory, Physics Unit, Faculty of Engineering and Natural Sciences, Tampere University, FI-33101 Tampere, Finland. [8] These authors contributed equally: Sophie Tomaz, Dongyu Wang. ✉email: matthieu.riva@ircelyon.univ-lyon1.fr

Atmospheric aerosols adversely affect human health[1] and have important climate effects. They impact Earth's radiative balance by directly interacting with light, or indirectly by acting as cloud condensation nuclei (CCN)[2,3]. Their effects on cloud formation and physical properties remain one of the most important uncertainties in climate models[3,4]. Up to 50% of global CCN formation could be attributed to new particle formation (NPF)[5–7]. Rapidly and widely formed, highly oxygenated organic molecules (HOMs)[8–11] have been identified to be critical for NPF and growth, and therefore for CCN formation[12–15].

HOM formation proceeds via autoxidation of the peroxy radical ($RO_2$), which undergoes intramolecular hydrogen abstraction (H-shift), yielding an alkyl radical (R) with a hydroperoxyl functional group (–OOH). Addition of $O_2$ to R produces a new $RO_2$ radical, which may undergo termination reactions or further autoxidation reactions[8,16,17]. Autoxidation can therefore yield a plethora of multifunctional products with low or extremely low saturation vapor pressures within seconds[10,18,19]. Studies have also reported rapid gas-phase formation of dimeric accretion products (i.e., organic peroxides, ROOR) produced from $RO_2 + RO_2$ reactions[8,16,20,21], which can initiate and contribute to NPF[22] and particle growth[20,23]. However, little is known about the molecular structures of the gas-phase HOMs, including $RO_2$ and ROOR, making their formation pathways uncertain. HOMs' contribution to NPF is directly dependent on their volatility, which can vary by orders of magnitude for isomeric compounds with different structures and functional groups[24,25].

While quantum chemical calculations and online high-resolution mass spectrometry (MS) analyses using isotopic labeling and deuterium-hydrogen exchange have provided valuable information on HOMs, direct assessment of their molecular structures remains elusive[26–29]. Existing online tandem MS (MS/MS) studies on low-volatility compounds are restricted to the aerosol phase[30–32]. Offline analysis requires sample collection and extensive sample preparation[33], which can be prone to artifacts from chemical degradation and contamination[34]. Nevertheless, MS/MS measurements in the particle phase[30,35–37] have provided precious insights into the structure of stable compounds but also unstable gas-phase precursors, for example the identification of stable ester dimers produced from α-pinene ozonolysis[36,38]. α-Pinene is one of the most abundant biogenic volatile organic compounds in the atmosphere, with global emission reaching up to 66.1 Tg yr$^{-1}$, and one of the key precursors for organic aerosol[39]. Similarly, the ozonolysis of limonene (structural isomer of α-pinene) is also expected to play a major role in NPF, despite the lower emission rate (11.4 Tg yr$^{-1}$), due to its greater reactivity toward ozone and a higher HOM yield than α-pinene[10]. In addition, limonene is found in cleaning and personal care products, and might be a significant source for indoor secondary organic aerosols (SOAs)[40–43]. Limonene has a single cyclic structure as opposed to the bicyclic structure of α-pinene, making it a model compound for the elucidation of monoterpene oxidation mechanisms due to the comparatively simple product structures.

In this study, we report the in-situ structural characterization of gas-phase HOMs, $RO_2$ radicals, and dimeric products of α-pinene and limonene ozonolysis based on online MS/MS using an ultrahigh-resolution Orbitrap mass spectrometer equipped with a $NO_3^-$ chemical ionization (CI) source[44,45]. Using online MS/MS data, aided by known structure-specific fragmentation patterns from existing offline MS/MS literature, we are able to infer the most plausible isomers of different compounds. We propose the most plausible oxidation product structures, autoxidation mechanisms, and dimer formation pathways based on the observed MS/MS product ions and neutral losses for the two monoterpene precursors.

## Results

**Online MS/MS**. Representative mass spectra of limonene and α-pinene ozonolysis products are shown in Supplementary Fig. 1. Online MS/MS was performed on $NO_3^-$ adducts of $C_{10}H_{14-16}O_{6-10}$ monomers, $C_{10}H_{15}O_{6,8,10}$ $RO_2$ radicals, and $C_{18-20}H_{28-34}O_{6-16}$ dimers at different collision energies (normalized collision energy (NCE) from 2 to 10). Overall, as shown for $C_{10}H_{14}O_7NO_3^-$ and $C_{20}H_{30}O_{14}NO_3^-$ in Supplementary Figs. 2 and 3, respectively, MS/MS spectrum exhibits a higher signal intensity corresponding to the precursor ion at lower collision energies, while product ions and $NO_3^-$ (declustered CI reagent ion) dominate at higher collision energies. In general, dimers appear to bind more strongly to $NO_3^-$ than do monomers (Supplementary Figs. 2c and 3c)[44]. In Fig. 1, we present an overview of most important neutral losses and product ions observed for HOMs. We infer the neutral loss pattern from the precursor and product ion formula (Fig. 1b). For closed-shell molecules, we observe a common fragmentation pattern involving $HNO_3$ loss, which is rare or insignificant for $RO_2$ radicals (Fig. 1a). We report the fragmentation patterns of $RO_2$ radicals (e.g., $C_{10}H_{15}O_{8,10}$), all of which undergo $O_2$ loss, corresponding to the loss of the peroxy functional group, as shown in Fig. 1a. $NO_3^-$ product ion is observed in all MS/MS spectra. While $RO_2$ radicals and closed-shell molecules share some common MS/MS fragments, predominately $C_{3–6}$ ions (Fig. 1a), they exhibit distinct neutral loss patterns and share little spectral similarity (Supplementary Fig. 4b). In general, compounds produced from the same (VOC + $O_3$) reactions share spectral similarity with each other (Fig. 1b), which is also reflected in the unsupervised agglomerative hierarchical clustering results (Supplementary Fig. 4a). As discussed below, the neutral loss patterns such as OH, $HO_2$, $CH_3O_2$, and $C_3H_6O$ observed for $RO_2$ radicals (Fig. 1a) are indicative of specific functional groups.

**Autoxidation mechanism**. An overview of limonene ozonolysis mechanism can be found in Fig. 2 and Supplementary Fig. 5. Briefly, limonene ozonolysis predominantly occurs by ozone addition onto the endocyclic double bond, forming an unstable primary ozonide, followed by the formation of two distinct Criegee intermediates[46–49]. The major Criegee intermediate can evolve into $RO_2$ radicals, $C_{10}H_{15}O_4$, by two main channels: A and B (Supplementary Fig. 5), which can undergo further autoxidation via intramolecular H-shifts followed by $O_2$ addition, forming more oxidized $RO_2$ radicals (i.e., $C_{10}H_{15}O_{6,8,10}$)[17,47,48]. Earlier studies have showed that aldehydic H-shift is rapid enough (i.e., $10^{-2}$ to $10$ s$^{-1}$) to be competitive against bimolecular reactions[50–53]. Thus, for limonene, the aldehydic H-shift is anticipated to be a major formation pathway for $C_{10}H_{15}O_6$ via a 1,9- or a 1,7 H-shift for A and B routes, respectively (Supplementary Fig. 5). We note that while the formation of A/B 4 via endocyclization is also expected to be fast[21,54], the corresponding fragmentation patterns were not observed in our MS/MS spectra. Figure 2 and Supplementary Fig. 5 labels possible $C_{10}H_{15}O_8$ radicals from both routes as A1–A7 and B1–B7, respectively. It is possible that some $RO_2$ isomers may exist to varying extents. Nonetheless, offline ultra-performance liquid chromatography–electrospray ionization–tandem mass spectrometry (UPLC–ESI–MS/MS) analyses show that a limited number of isomers dominate the $RO_2$ termination products (Supplementary Fig. 6), consistent with previous studies using ion mobility MS[55]. These prior studies suggest that a limited number of isomers are formed from the oxidation of monoterpenes. With this in mind, we demonstrate how MS/MS analyses can be utilized to constrain the most probable isomers and the corresponding H-shift pathways

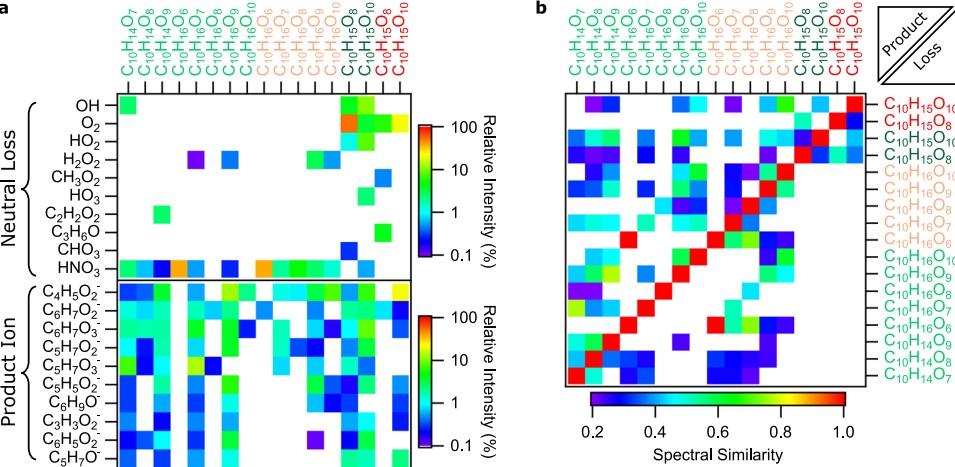

**Fig. 1 Common MS/MS product ions and neutral losses observed. a** Selected neutral loss patterns deduced from the formulae of the parent and product ions, and the ten most common product ions observed. **b** Pair-wise spectral similarities in MS/MS spectra. Corresponding MS/MS data were obtained at NCE 2 for $NO_3^-$ adducts of limonene and α-pinene ozonolysis products. Parent molecular formulae are shown on the upper x-axis with labels for closed-shell limonene and α-pinene ozonolysis products in light green and light red, respectively. Labels for limonene and α-pinene $RO_2$ radical are in dark green and dark red, respectively. Inferred neutral losses and observed product ions with relative intensity ≥0.1% are shown in log color in **a**. The upper triangle and lower triangle areas in **b** represent the cosine similarity between the observed product ions or neutral losses, respectively, among the MS/MS spectra. For clarity, only pairs with cosine similarity ≥0.2 are shown in linear color scale. Parent molecular formulae are also shown on the right y-axis in **b**.

given the observed product ions, neutral losses, and known fragmentation mechanisms.

Figure 3a shows that the radical $C_{10}H_{15}O_8NO_3^{\bullet-}$ ($m/z$ 325.0644) first undergoes $O_2$ elimination during fragmentation, forming the alkyl radical-anion $C_{10}H_{15}O_6NO_3^{\bullet-}$ ($m/z$ 293.0748), followed by the simultaneous loss of $HNO_3$ and OH, yielding $C_{10}H_{13}O_5^-$ ($m/z$ 213.0766) (Fig. 3c). This can only occur if the carbon adjacent to the alkyl radical is not quaternary, therefore excluding structures A/B 4, 5, 6 and B7. The $H_2O$ elimination from the resulting $C_{10}H_{13}O_5^-$ to form $C_{10}H_{11}O_4^-$ ($m/z$ 195.0667) likely occurs on the hydroperoxide functional group (Fig. 3c), a known negative ion fragmentation pathway[56]. The $C_9H_{11}O_2^-$ ($m/z$ 151.0764) product ion results from $C_{10}H_{11}O_4^-$ via a charge migration elimination of $CO_2$, which is indicative of a carboxylic functional group, related to a peroxy acid moiety formed from the aldehydic H-shift. Finally, the formation of $C_{10}H_{14}O_7NO_3^-$ ($m/z$ 308.0616) from $C_{10}H_{15}O_8NO_3^{\bullet-}$ through an intramolecular H-abstraction from the peroxy radical moiety, followed by OH elimination and ketone formation, is not possible for tertiary alkylperoxy radicals such as A/B 3. Based on the MS/MS analysis, A/B 1 and 2 and A7 from 1,4, 1,6, or 1,7 H-shift emerge as the most plausible structures for $C_{10}H_{15}O_8$, under the assumption that the A/B $O_6$ peroxy radical structures were the precursors of the $O_8$ radicals. Additionally, if we consider rapid H-shift scrambling (scrb)[57], B1-3 and 7-scrb structures (with B7-scrb corresponding to A7) may also be plausible structures for $C_{10}H_{15}O_8$, as these structures can accommodate the observed MS/MS fragmentation pathways.

Assuming A/B 1,2 and A7 as the main structures of $C_{10}H_{15}O_8$, several $C_{10}H_{15}O_{10}$ structures are possible (Fig. 1 and Supplementary Figs. 7 and 8). In the MS/MS spectrum of $C_{10}H_{15}O_{10}NO_3^{\bullet-}$ ($m/z$ 357.0543, Fig. 3b), the product ion $C_{10}H_{14}O_9NO_3^-$ ($m/z$ 340.0515) is likely formed via OH elimination similar to that depicted in Fig. 3d, which is not possible for tertiary alkyl peroxy radicals, thereby eliminating several potential candidates (A1–6, A2–5, A7–1, A7–5, B1–6, and B2–6). The formation of $C_{10}H_{14}O_8NO_3^-$ ($m/z$ 324.0565) from $C_{10}H_{15}O_{10}NO_3^{\bullet-}$ via $HO_2$ elimination occurs if the carbon next to the $RO_2$ radical is not quaternary, limiting the potential precursor structures to A/B 1–1, A/B 1–2, A/B 2–1, and A 7–2. Following $O_2$ elimination from $C_{10}H_{15}O_{10}NO_3^{\bullet-}$, the subsequent

$CHO_3$ elimination from $C_{10}H_{15}O_8NO_3^{\bullet-}$ to form $C_9H_{14}O_5NO_3^-$ ($m/z$ 264.0725) can follow a radical recombination mechanism described in Fig. 3d. Such recombination is possible if the radical is near a peroxy functional group, excluding the A/B 1–2 structures. Therefore, A/B 1–1, 2–1, and A 7−2 are likely the only candidates for $C_{10}H_{15}O_{10}$, corresponding to a 1,5 and 1,6 H-shift from saturated carbon atoms, for which all observed loss processes are feasible. Lastly, we note that if H-scrambling between an $RO_2$ site and hydroperoxide functional groups occurs in $C_{10}H_{15}O_8$ and $C_{10}H_{15}O_{10}$ radicals, up to 14 additional potential structures may also explain the observed $O_{10}$ MS/MS fragmentation pattern.

As previously described and discussed in the Supplementary Information (Supplementary Figs. 9–11), the MS/MS spectrum of $C_{10}H_{15}O_8$ from α-pinene ozonolysis (Supplementary Fig. 10a) exhibits similar fragmentations (i.e., $O_2$ and $CO_2$ eliminations corresponding to the $RO_2$ radical moiety and the peroxy acid functional group, respectively) but also some notable differences compared to products from limonene oxidation. Previous works investigated the fragmentation of compounds bearing a peroxy acid functional group and described a common fragmentation loss via $CH_2O_2$ elimination through a two-step fragmentation process ($H_2O + CO$)[58,59], which is not observed for either α-pinene or limonene. In addition, the MS/MS spectrum of α-pinene $C_{10}H_{15}O_8^-$ $RO_2$ radical displays two distinct fragmentation losses, $CH_3O_2$ and $C_3H_6O$, which are indicative of the methyl group specific to α-pinene $RO_2$, i.e., quaternary carbon bonded to two methyl groups (Supplementary Fig. 9, highlighted in blue), which is not present in the case of limonene $RO_2$.

**Gas-phase dimer formation mechanism.** The reaction rate coefficient between $RO_2$ radicals can range from $10^{-17}$ to $10^{-11}$ $cm^3$ molecule$^{-1}$ s$^{-1}$ [60], with recent studies reporting values on the order of $10^{-10}$ $cm^3$ molecule$^{-1}$ s$^{-1}$ for selected HOM-$RO_2$ [21,61]. While kinetic studies show that the $RO_2 + RO_2 \rightarrow ROOR + O_2$ reaction rate coefficients generally increase with the $RO_2$ oxygen content[62], further enhanced by intramolecular hydrogen bonds that may help with transition state stabilization[28], the contribution of different $RO_2$ radicals to dimer formation remains

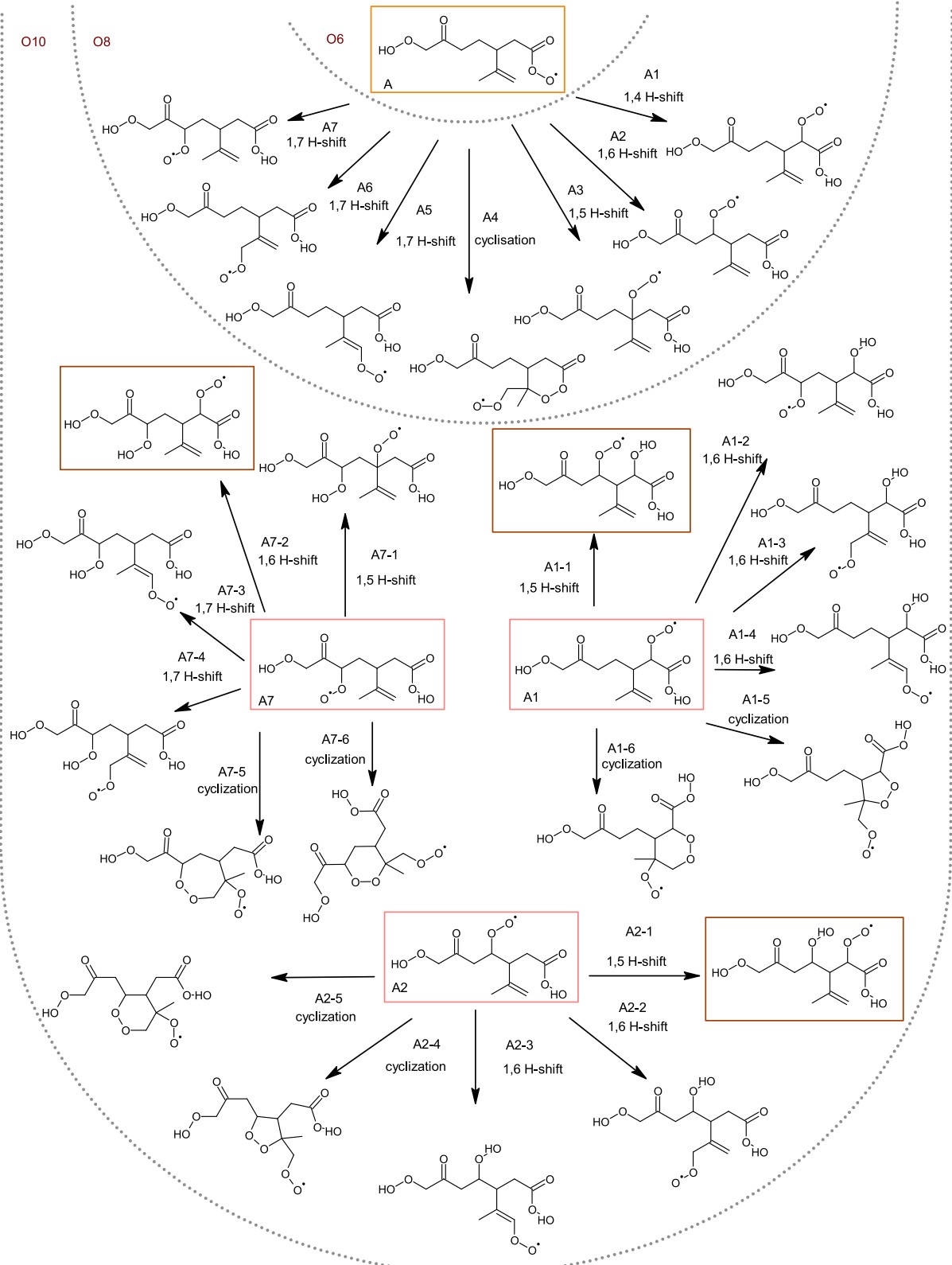

**Fig. 2 Proposed limonene autooxidation pathways and products.** Potential structures and proposed formation mechanisms of the $O_6$, $O_8$, and $O_{10}$ $RO_2$ radicals from the A route of the ozonolysis of limonene. Most plausible identified structures are boxed in orange, pink, or brown for $O_6$, $O_8$, and $O_{10}$ $RO_2$ radicals, respectively.

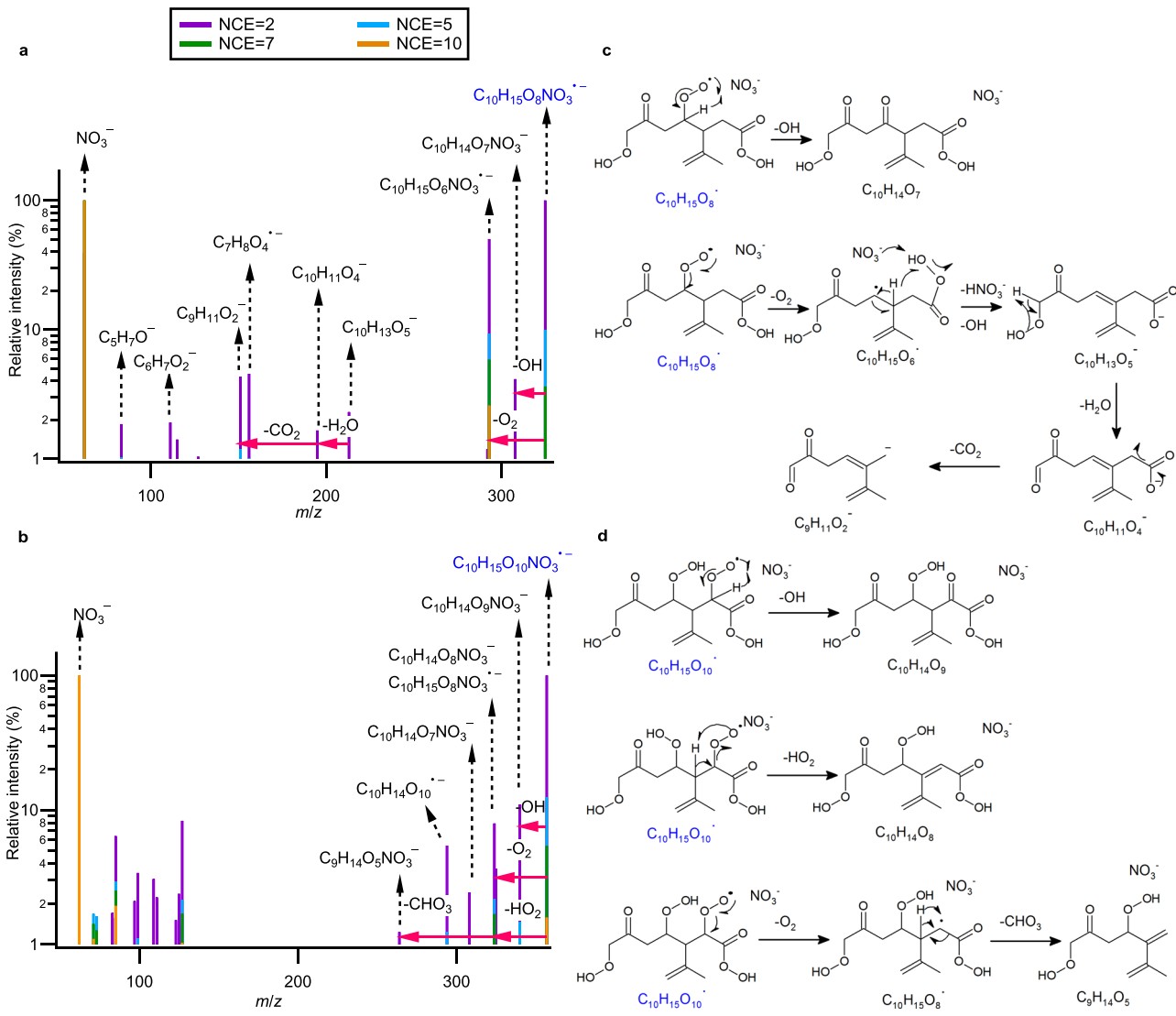

**Fig. 3 MS/MS spectra of two limonene ozonolysis RO₂ radicals at various NCE (2–10). a** MS/MS spectra of $C_{10}H_{15}O_8NO_3^{•-}$. **b** MS/MS spectra of $C_{10}H_{15}O_{10}NO_3^{•-}$. **c** MS/MS fragmentation routes to product ions, $C_{10}H_{14}O_7NO_3^-$ and $C_9H_{11}O_2^-$ from the A2 structure of $C_{10}H_{15}O_8NO_3^{•-}$ as depicted in Fig. 2. **d** MS/MS fragmentation route to product ions, $C_{10}H_{14}O_9NO_3^-$, $C_{10}H_{14}O_8NO_3^-$, and $C_9H_{14}O_5NO_3^-$, from the structure B2-1 of $C_{10}H_{15}O_{10}NO_3^{•-}$ as depicted in Supplementary Fig. 7. The parent radical anions are highlighted in blue.

unknown. Online MS/MS analysis provides insights into the dimer structure and formation, as shown for $C_{20}H_{30}O_{12}NO_3^-$ (m/z 524.1608) and $C_{20}H_{30}O_{14}NO_3^-$ (m/z 556.1520) in Fig. 4 for limonene ozonolysis and in Supplementary Fig. 12 for α-pinene ozonolysis. $C_{20}H_{30}O_{12}$ can be formed from reactions of $O_4$ with $O_{10}$ or $O_6$ with $O_8$ peroxy radicals; $C_{20}H_{30}O_{14}$ can be formed from reactions of $O_8$ with $O_8$, or $O_6$ with $O_{10}$ peroxy radicals. As shown in Fig. 4a and b, both limonene ozonolysis dimers produce the $C_{10}H_{14}O_7NO_3^-$ (m/z 308.0617) product ion, which is likely formed from the cleavage of the peroxide (i.e., RO–OR bond) into an alkoxy radical (RO) combined with H-exchange, as illustrated in Fig. 4c. The presence of $C_{10}H_{14}O_7NO_3^-$ suggests that $C_{10}H_{15}O_8$ may be the main precursor for $C_{20}H_{30}O_{12}$ and $C_{20}H_{30}O_{14}$. This is also supported by the absence of $O_9$ fragments and the presence of $C_{10}H_{13}O_7^-$ (m/z 245.0663). This is further reinforced by hierarchical clustering analyses (Supplementary Fig. 13), which consistently group $C_{10}H_{16}O_8$ (termination product of $C_{10}H_{15}O_8$), $C_{20}H_{30}O_{12}$, and $C_{20}H_{30}O_{14}$ together based on their MS/MS spectral similarity. The hierarchical clustering analysis also reveals that the MS/MS spectrum of $C_{10}H_{16}O_7NO_3^-$ is

distinct from that of $C_{20}H_{30}O_{12}NO_3^-$ and $C_{20}H_{30}O_{14}NO_3^-$ (Supplementary Fig. 13). Similarly, the importance of the $O_6$-radical (i.e., $C_{10}H_{15}O_6$) in the formation of the $C_{20}H_{30}O_{12}$ is supported by the detection of $C_{10}H_{15}O_5^-$ (m/z 215.0921). While $C_{20}H_{30}O_{14}NO_3^-$ produces a $C_{10}H_{16}O_7NO_3^-$ (m/z 310.0773) product ion (Fig. 4b), $C_{20}H_{30}O_{12}NO_3^-$ does not (Fig. 4a). This difference indicates that $C_{10}H_{15}O_6$ is an acylperoxy radical, which is consistent with the structure proposed in Fig. 4d. As a result, only one RO–OR cleavage pathway for $C_{20}H_{30}O_{12}NO_3^-$ is possible with the H-exchange only taking place in one side of the peroxide, in contrast to $C_{20}H_{30}O_{14}NO_3^-$ (Fig. 4c). This result also supports the structures A/B proposed in Fig. 2 and Supplementary Fig. 5.

The fragmentation patterns of α-pinene dimers are distinct from those of limonene, suggesting a considerable difference in the reactivity of their $RO_2$ radicals (as discussed below). For example, the reaction between $C_{10}H_{15}O_{10}$ and $C_{10}H_{15}O_4$ $RO_2$ radicals clearly contributes to the formation of $C_{20}H_{30}O_{12}$ dimer from α-pinene ozonolysis, according to the presence of $C_{10}H_{16}O_9NO_3^-$ (m/z 342.0676) and $C_{10}H_{15}O_9^-$ (m/z 279.0722)

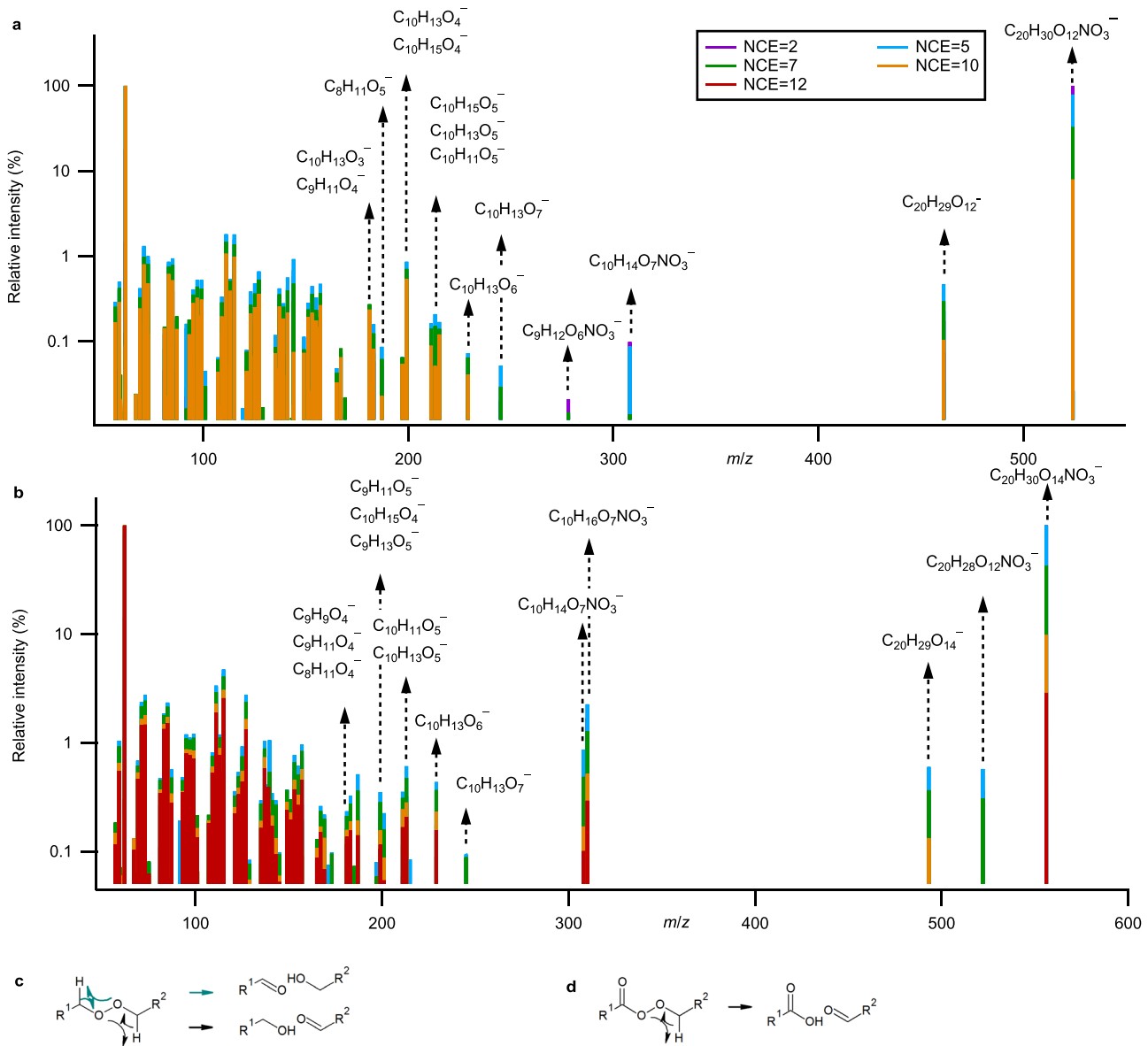

**Fig. 4 MS/MS spectra of two limonene ozonolysis dimers at various NCE (2–10). a** MS/MS spectra of $C_{20}H_{30}O_{12}NO_3^-$. **b** MS/MS spectra of $C_{20}H_{30}O_{14}NO_3^-$. **c** MS/MS fragmentation mechanism of RO–OR for dimers produced from the reaction involving two peroxy radical precursors, which can proceed via either of the black or green pathways. **d** MS/MS fragmentation mechanism of RO–OR for dimers produced from the reaction involving an acylperoxy radical and a peroxy radical, where only one pathway is available.

product ions in the MS/MS spectrum, as shown in Supplementary Fig. 12. The $C_{10}H_{15}O_6^-$ ($m/z$ 231.0874) product ion observed in the MS/MS spectrum might suggest that $RO_2$ radicals other than $O_4$ and $O_{10}$ may contribute to the formation of dimer, although this product ion may have also originated from further fragmentation of a larger product ion. Hence, reactions between $C_{10}H_{15}O_6$ and $C_{10}H_{15}O_8$ radicals may also contribute to the formation of $C_{20}H_{30}O_{12}$ dimers, though the corresponding RO–OR product ions, e.g., $C_{10}H_{16}O_5NO_3^-$ and $C_{10}H_{16}O_7NO_3^-$, were not observed under our experimental conditions. The greater importance of the $O_4$ α-pinene $RO_2$ radical compared to the $O_4$ limonene $RO_2$ radical is also reflected in the dimer concentration distributions (Supplementary Fig. 1d), with less oxygenated dimers being more abundant during α-pinene ozonolysis ($C_{20}H_{30}O_{10}$ and $C_{20}H_{30}O_{12}$) than during limonene ozonolysis ($C_{20}H_{30}O_{12}$ and $C_{20}H_{30}O_{14}$). Ozonolysis experiment with varying α-pinene concentration further indicates that the $O_4$ α-pinene $RO_2$ radical is a key participant in the formation of

$C_{20}H_{30}O_8$ (with $C_{10}H_{15}O_6$), $C_{20}H_{30}O_{10}$ (with $C_{10}H_{15}O_8$), and $C_{20}H_{30}O_{12}$ (with $C_{10}H_{15}O_{10}$) dimers (see Supplementary Figs. 15, 16 and Supplementary Information).

## Discussion

We have shown that the structures A/B-1, A/B-2, and A-7 may be potential $RO_2$ radicals formed during limonene autoxidation. The formation of these structures requires 1,6-, 1,7-, and 1,4-H shifts, respectively. For A1 and B1, the required 1,4 H-shifts are typically thought to be slow, due to steric hindrance[10], but theoretical calculations show that 1,4-H-aldehydic migration has significantly lower energy barriers than other H-shift pathways on unsubstituted aliphatic carbons[63]. The presence of adjacent functional groups such as alkyl, carbonyls, hydroxyl, hydroperoxyl, and ether can further enhance H-abstraction[10,64]. Unlike 1,4 H-shifts, 1,6 and 1,7 H shifts are not expected to cause steric strains, though H-abstraction from a saturated carbon atom is

likely slow[63]. Structures resulting from a saturated carbon H-shift were shown to be good candidates and could explain MS/MS fragmentation for both limonene and α-pinene peroxy radicals in our work, despite their structural differences. Our data also highlight the presence of peroxy acid functional groups for both limonene and α-pinene peroxy radicals, which has also been suggested for other VOCs[16,65]. This is further supported by off-line UHPLC–ESI–MS/MS analyses of $C_{10}H_{16}O_{7-8}$ and $C_{10}H_{16}O_{5-6}$ (termination products of the $O_8$ and $O_6$ peroxy radicals, respectively), largely detected in the negative ion mode (Supplementary Fig. 6), which is consistent with the presence of an acidic hydrogen from the peroxy acid functional group [58,59].

When applied to gaseous dimers, online MS/MS reveals unique RO–OR dimer fragmentation pathways that help identifying main $RO_2$ radical precursors, which is further supported by hierarchical clustering analyses. Our results show that HOM dimer formation during limonene ozonolysis is driven by the $O_8$ peroxy radical (Fig. 4 and Supplementary Figs. 14 and 16), whereas the $O_4$ peroxy radical appears to be a key component for dimer formation during α-pinene ozonolysis (Supplementary Figs. 12, 14, and 16). Consequently, dimers produced from α-pinene ozonolysis were on averaged less oxidized than those produced from limonene ozonolysis (Supplementary Fig. 1d) despite seemingly similar $RO_2$ radical distribution ($O_8 > O_{10} > O_6$, Supplementary Fig. 1c), with potential implications of reduced NPF due to the higher vapor pressure of α-pinene dimers[22]. Furthermore, our results show that the $RO_2 + RO_2$ reaction rate coefficient is not only driven by the degree of oxidation[61] (Supplementary Fig. 16) but also by the structure/reactivity of each $RO_2$ radicals. Overall, our findings advance the understanding of atmospheric radical chemistry, which can help constraining model representation of autoxidation pathways and dimer formation kinetics. The online approach employed here can be readily applied to, and is beneficial for, the investigation of short-lived or labile organic compounds in the gas phase present in low concentrations, demonstrably organic radicals that are otherwise inaccessible by offline techniques.

## Methods

**General information and procedures**. Ozonolysis experiments were performed in a 18-liter Pyrex glass flow tube reactor (12 cm i.d. × 158 cm length) at room temperature[66]. Continuous injection of volatile organic compound (VOC) precursors, (+)-limonene (Sigma-Aldrich, 97% purity) or (+)-α-pinene (Sigma-Aldrich, ≥99%), ozone, and flow reactor carrier gas was regulated using mass flow controllers (MFC, Bronkhorst). Dry synthetic air (80:20 $N_2:O_2$) was used as the carrier gas for the flow tube and VOC injection. Total flow rate is kept at 21 L min$^{-1}$ and the total reaction time is 59.1 s. Input VOC concentrations, ranging from 45 to 227 ppbv for limonene and 214 to 749 ppbv for α-pinene, are estimated based on MFC settings and VOC evaporator temperature (5 °C). Ozone was generated via dielectric barrier discharge and was monitored with a Thermo 49C analyzer, ranging from 20 to 40 ppbv. Negligible particle formation (<100 cm$^{-3}$) was observed by a condensation particle counter (TSI CPC 3772) during all experiments. A Q Exactive Orbitrap mass spectrometer (Thermo Scientific, US) coupled to an atmospheric pressure CI inlet[44] was used for online analysis of ozonolysis products. The Orbitrap has been used with an automatic gain control (AGC) and maximum injection time set to $1 \times 10^5$ and 3000 ms, respectively. The mass resolution is set to 140,000 at m/z 200. The CI reagent nitrate ion ($NO_3^-$) was generated from a nitric acid solution (Sigma-Aldrich, 65% purity) continuously flushed with pure $N_2$ (10 mL min$^{-1}$) and ionized with a soft X-ray photoionizer (Hamamatsu, L9491). The CI inlet total flow and sheath flow rates were set to 36 L min$^{-1}$ and 34 L min$^{-1}$, respectively. Further details of instrument setting and experimental conditions can be found in the Supplementary information.

**Online tandem MS**. Higher energy collision dissociation (HCD) was used to obtain targeted MS/MS spectra using a quadrupole ion isolation window width of 0.4 Da. No in-source collision-induced dissociation (CID) energy was applied. The HCD cell is located after the quadrupole capable of mass isolation within ±0.4 Da. The selectivity of the $NO_3^-$ detection scheme toward acidic and highly oxidized compounds ensures that a single parent ion is selected for structural characterization using MS/MS acquisitions for the $RO_2$ radicals and dimeric products discussed in the main text within the mass isolation window. In the case that more

than one parent ion is isolated, the dominant parent ion is at least an order of magnitude more abundant than the other parent ion(s) in most cases. An NCE of 2 was applied to $RO_2$ and closed-shell monomers. An NCE value of 5 was applied to dimers, as well as to select $RO_2$ and monomers for comparison. Systematic ramping of NCE from 2–10 was also performed for a few targeted compounds. The Orbitrap was mass-calibrated using an aqueous sodium acetate solution (2 mM, Aldrich, >99% purity). During online CI-measurements, the drift in mass accuracy was negligible (i.e., <2 p.p.m.) based on the reported m/z of the CI reagent ions ($NO_3^-$ and $HNO_3NO_3^-$) and m/z values from MS/MS parent ions, which is within the specifications of the Orbitrap. Peak identification and assignment were performed using XCalibur 4.1 (Thermo Scientific). Identification of parent ion formula was constrained based on known oxidation chemistry and elemental composition, i.e., ions are assumed to contain only carbon, hydrogen, and oxygen atoms with up to two nitrogen atoms to account for adduction with $NO_3^-$ or $HNO_3NO_3^-$. Identification of MS/MS product ion is constrained by the parent ion elemental composition. Product ion intensities were normalized to the maximum ion intensity unless stated otherwise. A 5 p.p.m. m/z tolerance and a 0.1% relative intensity threshold were applied to refine the product ion lists. Agglomerative hierarchical clustering of the processed MS/MS spectra was performed in Spyder 3.3.2 using complete linkage of cosine distance of square root-transformed product ion relative intensities as recommended in the literature[67], excluding $NO_3^-$ product ions, whose relative contribution reflects more the dissociation energy of $NO_3^-$ adducts than the analyte structure or functionalization. The precursor ion signal is the most intense in all MS/MS spectra acquired at NCE ≤ 5 after excluding the $NO_3^-$ ion. Clustering analysis results based on observed product ions alone or together with inferred neutral losses are detailed in the Supplementary Information. Details of the offline analysis of monoterpene ozonolysis products using high-performance liquid chromatography with Orbitrap are found in the Supplementary Information.

## Data availability

The data supporting the finding of this study are available in this article and Supplementary information. Data presented in the main text are accessible via https://doi.org/10.5281/zenodo.4276954. Raw mass spectrometric data are available from the corresponding author on reasonable request.

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

## Acknowledgements
This work was financially supported by the University of Lyon through the Breakthrough grant WANTED, the French National program LEFE (Les Enveloppes Fluides et l'Environnement), the Academy of Finland (grants 317380 & 320094), the Swiss National Science Foundation (no. 200020_172602), the European Union's Horizon 2020 research and innovation program under Marie Skłodowska-Curie grant agreement (no. 690958 & no. 701647), ERC-StG COALA (no. 638703), and ERC-StG MAARvEL (no. 852161). M.E.M. is a Research Staff member from CONICET. The authors thank Frederic Bourgain for technical support.

## Author contributions
M.Riva conceived the work. S.T., D.W., N.Z., I.E.-H., and M.Riva designed the experiments and analyzed the data. S.T., D.W., N.Z., D.L., and M.Riva performed the flow tube experiments. F.B. and A.V. performed the offline analysis. S.T., D.W., I.E.-H., and M.Riva wrote the manuscript. N.Z., D.L., H.L., F.B., A.V., M.E.M., S.P., U.B., C.G., M.R., and M.E. commented on the manuscript.

## Competing interests
The authors declare no competing interests.

## Additional information

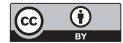

