## [Peer Review File · Nature Communications]

REVIEWER COMMENTS

Reviewer #1 (Remarks to the Author):

Referee: Magda Claeys

General comments:

This is a thoughtful study providing detailed insights into structures and reactivity of RO₂ radicals and dimeric products from monoterpene ozonolysis using online tandem mass spectrometry (MS/MS). As the authors write, the structure and formation mechanism of RO₂ radicals and dimeric products remain elusive. Online MS/MS analysis of gas-phase products using an Orbitrap analyzer is appropriate methodology to address this gap. Insight into the formation of RO₂ radicals and highly oxygenated molecules (HOMs) is indeed very important as it has been shown that HOMs are involved in formation and growth of new particles. Formation mechanisms involving unstable intermediates are difficult to formulate because unstable compounds cannot be chemically isolated and structurally characterized; however, the molecular structure of the gas-phase precursor (e.g., α -pinene), its known gas-phase chemistry, the molecular composition of unstable intermediates and the molecular structure of stable end products observed in the particle phase can provide crucial insights. The current study has an additional dimension and addresses the molecular structure (and not only the composition) of gas-phase unstable intermediates. This is extremely challenging but it has to be remembered that MS/MS analysis of unstable intermediates is only one of the analytical tools and can only provide partial structural information based on the molecular composition of specific neutral losses and product ions. In the current study, the MS/MS results have been interpreted in detail and appear logical. I fully agree with one of the conclusions drawn that tandem MS is a unique approach for the characterization of extremely reactive compounds such as organic radicals. A major comment relates to the conclusion drawn about dimer formation from α -pinene ozonolysis (see specific comment lines 232-235).

Specific comments:

Two gas-phase precursors have been selected for this study, i.e., limonene and α -pinene. The choice of α -pinene is fine as it has been documented that dimeric products related to α -pinene have been found in the particle phase of ambient environments and structurally elucidated. However, I am not aware that this is the case for limonene. I understand that limonene is useful as a model monoterpene before studying α -pinene, which has a more complex bicyclic structure. I suggest to give some additional motivation for first considering limonene.

Lines 70-73: I strongly believe that the structure elucidation of stable dimeric end products observed in the particle phase can also provide useful information on the structure of the unstable gas-phase precursors. In other fields of science, e.g., research on the arachidonic acid cascade, scientists have deduced the structure of the active highly oxygenated precursors (endoperoxides) based on the structures of the stable end products, e.g., prostaglandins and thromboxanes. In my opinion, it is not possible to deduce the structure of unstable dimeric HOMs without considering the well-established structures of particle-phase dimeric HOMs.

Lines 232-235: It is suggested that the O₄ peroxy radical is a key component for dimer formation from α -pinene ozonolysis and involves reactions between C₁₀H₁₅O₁₀ and C₁₀H₁₅O₄. This conclusion is made on the basis of RO₂ + RO₂ → ROOR + O₂ chemistry and the presence of C₁₀H₁₆O₉NO₃⁻ and C₁₀H₁₅O₉⁻ product ions in the MS/MS spectrum of C₂₀H₃₀O₁₂⁻ (Figure S12). However, examining this spectrum one can note another even more abundant product ion, i.e., C₁₀H₁₅O₆⁻, which leads me to conclude that the O₆ peroxy radical could also participate in the formation of the C₂₀H₃₀O₁₂ dimers. It is likely that the C₂₀H₃₀O₁₂ dimers are a mixture of products, where not only O₄ but also O₆ peroxy radicals play a key role in dimer formation.

Line 198: It is not clear what is meant by a quaternary carbon bonded to two methyl groups. Perhaps it is best to refer to a scheme.

Technical comments (mainly related to MS terminology):

Lines 91-92 and several places elsewhere: I recommend to follow the IUPAC guidelines for terms related to mass spectrometry. The term "fragment ion" is deprecated and not the recommended term (only to be used in the case of fragmentation in the electron ionization mode); the recommended term is "product ion". The IUPAC guidelines can be found in the following paper: Ref.: K. K. Murray, R. K. Boyd, M. N. Eberlin, G. J. Langley, L. Li, Y. Naito. Definitions of terms relating to mass spectrometry. IUPAC Recommendations 2013. Pure Appl. Chem., 85, 1515-1609, 2013.

Line 94-95: same comment as above: replace "charged fragment" by "product ions".

Line 101: MS/MS product ions

Line 116: observed product ions with relative abundance (Note: ions are abundant but ion signals are intense!).

Line 173: the product ion

Line 234: product ions in the MS/MS

Line 313: Product ion abundances were normalized to the maximum ion abundance

Line 315: 0.1% relative abundance

Lines 317-318: product ion relative abundances

Line 319: NO₃- product ions, ...

Line 320: The precursor ion is the most abundant ion

Line 322: observed product ions

Line 363: Kerminen, V.-M. et al.

References: journal abbreviations should be checked; titles of journal articles should not be capitalized.

The supplement should also be checked for correct MS terminology.

Reviewer #2 (Remarks to the Author):

The submitted work describes the results of CI-MS studies of oxidation products from selected natural hydrocarbons. The topic is important for the understanding of atmospheric particle formation and has been the subject of comparatively intensive research in recent years. The authors focus on two monoterpenes, which both play a special role in atmospheric particle formation. The experiments have been performed at a high level and provide interesting, undoubtedly publishable results about possible structures, reactivities and especially fragmentation paths of the observed ions. The manuscript is well written, the main references are cited and justified in relation to the conclusions drawn. My only point of criticism is to what extent the authors meet their own claim, namely "... (to) present the first experimental elucidation of RO₂ and dimer structures....". What the authors do show, however, are potential structures and suggestions of formation pathways that represent only the 'most plausible' structures of the fragments. In this respect the question arises if the manuscript represents an advance in understanding likely to influence thinking in the field or if it is simply a very good publication for e.g. a mass spectrometric journal.

Reviewer #3 (Remarks to the Author):

Dear Editor, dear authors,

I thoroughly read the article Structures and Reactivity of Peroxy Radicals and Dimeric Products Revealed by Online Tandem Mass Spectrometry by Sophie Tomaz, Dongyu Wang et al submitted to Nature Communications (NCOMMS-20-29025-T).

The study presents new and certainly relevant data on the structure analysis of highly oxygenated organic molecules (HOMs), which contribute to the formation and growth of atmospheric particles. Specifically, the authors report new results on the mechanism and reaction pathways of organic peroxy radicals (RO₂) which are extremely reactive compounds, that are of vital importance for the atmospheric degradation of volatile hydrocarbons. Ultimately, highly oxygenated organic molecules (HOMs) are formed, including low-volatility ROOR dimers generated by bimolecular RO₂

+ RO₂ reactions. In the report, a promising experimental approach by online tandem mass spectrometry is presented to investigate the ions of interest in the gas-phase. The authors claim that their analytical strategy allows to constrain the structures and formation pathway of several HOM-RO₂ radicals and dimers produced from monoterpene ozonolysis, a prominent atmospheric oxidation process.

At this point it is important to note that the authors report extended mechanisms on the basis of their CID experiments of the NO₃⁻ adduct molecular ions but the majority of the structures shown are: tentative or most probable isomers – which is ok and correct as the ion structures are not proven besides their composition meeting the accurate ion mass ... This problem is certainly a known one for this kind of studies and does not diminish the value of the work, which is in my eyes still respectable and well conducted. However, I strongly recommend to tone down the wording in the abstract and the intro part. There is no final and exhaustive evidence for the ion structures found in the paper.

The recently introduced analytical technique comprises a NO₃⁻ - chemical ionization (CI) source coupled to an ultrahigh-resolution Orbitrap mass spectrometer. In addition to its high mass resolution (140 000 at m/z 200 for MS and 70 000 at m/z 200 for MS/MS analysis at FWHM) and fast time resolution (10s), the CI-Orbitrap is also capable of performing online MS/MS analysis of compounds at sub-parts per trillion by volume (pptv) level. This powerful set-up was already introduced and reported in two previous papers by the laboratory (see refs 38 and 39).

In this regard I strongly recommend to report only accurate ion masses with 5 decimal places (!!)

when these values are justified by minimal experimental errors of the respective accurate ion mass measurement. This mandatory information is missing. Without this information these values are not acceptable. High resolution capability alone is not the key information needed here – It is important to also report how accurate and precise your measurements are.

In conclusion I strongly recommend to clarify the wording on the ion structure assumptions in the discussions of the mechanisms. Additionally, the lacking infos needed to complete the experimental part on the accurate ion mass determinations have to be given before a final decision on the publication can be drawn.

Responses to Reviewer Comments

We thank the reviewers for their comments and suggestions, reprinted below in blue. Our point-by-point responses are shown below in black. Changes to the manuscript are displayed in green.

Reviewer #1 (Remarks to the Author):

Referee: Magda Claeys

General comments:

This is a thoughtful study providing detailed insights into structures and reactivity of RO₂ radicals and dimeric products from monoterpene ozonolysis using online tandem mass spectrometry (MS/MS). As the authors write, the structure and formation mechanism of RO₂ radicals and dimeric products remain elusive. Online MS/MS analysis of gas-phase products using an Orbitrap analyzer is appropriate methodology to address this gap. Insight into the formation of RO₂ radicals and highly oxygenated molecules (HOMs) is indeed very important as it has been shown that HOMs are involved in formation and growth of new particles. Formation mechanisms involving unstable intermediates are difficult to formulate because unstable compounds cannot be chemically isolated and structurally characterized; however, the molecular structure of the gas-phase precursor (e.g., α -pinene), its known gas-phase chemistry, the molecular composition of unstable intermediates and the molecular structure of stable end products observed in the particle phase can provide crucial insights. The current study has an additional dimension and addresses the molecular structure (and not only the composition) of gas-phase unstable intermediates. This is extremely challenging but it has to be remembered that MS/MS analysis of unstable intermediates is only one of the analytical tools and can only provide partial structural information based on the molecular composition of specific neutral losses and product ions. In the current study, the MS/MS results have been interpreted in detail and appear logical. I fully agree with one of the conclusions drawn that tandem MS is a unique approach for the characterization of extremely reactive compounds such as organic radicals. A major comment relates to the conclusion drawn about dimer formation from α -pinene ozonolysis (see specific comment lines 232-235).

-We thank the reviewer for the careful consideration of our study.

Specific comments:

Two gas-phase precursors have been selected for this study, i.e., limonene and α -pinene. The choice of α -pinene is fine as it has been documented that dimeric products related to α -pinene have been found in the particle phase of ambient environments and structurally elucidated. However, I am not aware that this the case for limonene. I understand that limonene is useful as a model monoterpene before studying α -pinene, which has a more complex bicyclic structure. I suggest to give some additional motivation for first considering limonene.

-We have added the following motivations about our choice of studying limonene. The end of the introduction section now reads as follows:

Page 5, lines 81-92: “In this work, this technique is applied to characterize the structure of closed-shell products and radicals produced from the oxidation of two monoterpenes, α -pinene and limonene. α -Pinene was chosen due to its global emission reaching up to 66.1 Tg yr⁻¹ and its importance in the formation of organic aerosol.⁴¹ Despite the lower emission rate of limonene (11.4 Tg yr⁻¹), limonene has a greater reactivity towards ozone and a higher HOM yield than α -pinene.¹⁰ As a result, limonene ozonolysis is expected to play a major role in new particle formation. In addition, limonene is found in cleaning and personal care products, and might be a significant source for indoor secondary organic aerosols (SOA).⁴¹⁻⁴⁴ Finally, limonene has a single cyclic structure as opposed to the bicyclic structure of α -pinene, which simplifies the interpretation of the MS/MS fragmentation patterns. We propose the most plausible oxidation product structures, autoxidation mechanism, and dimer

formation pathways based on the observed MS/MS product ions and neutral losses for the two monoterpene precursors.”

Lines 70-73: I strongly believe that the structure elucidation of stable dimeric end products observed in the particle phase can also provide useful information on the structure of the unstable gas-phase precursors. In other fields of science, e.g., research on the arachidonic acid cascade, scientists have deduced the structure of the active highly oxygenated precursors (endoperoxides) based on the structures of the stable end products, e.g., prostaglandins and thromboxanes. In my opinion, it is not possible to deduce the structure of unstable dimeric HOMs without considering the well-established structures of particle-phase dimeric HOMs.

-We agree with the reviewer and we would like to underline that we used existing literature based on offline techniques (i.e., filter and/or PILS samples characterized by LC/ESI-MS) to infer the structures of the reactive species studied in this work. We have made changes to better highlight the usefulness of previous investigations.

Page 4, lines 64-73: “Offline measurements in the particle phase utilizing tandem MS (MS/MS)³⁰⁻³³ experiments have provided precious insights into the structure of stable compounds but also unstable gas-phase precursors, for example the identification of stable ester dimers produced from α -pinene ozonolysis.^{32,34} In this study, we have used existing literature based on offline techniques to support the interpretations of the MS/MS fragmentation patterns for deriving structural information. However, offline analysis requires sample collection and extensive sample preparation,³⁵ which can be prone to artifacts from chemical degradation and contamination.³⁶ Existing online MS/MS studies on low-volatility compounds are also restricted to the aerosol phase^{31,37,38} As a result, the chemical structures and physicochemical properties of HOMs, RO₂ radicals and gas-phase dimers remain unknown.”

Lines 232-235: It is suggested that the O₄ peroxy radical is a key component for dimer formation from α -pinene ozonolysis and involves reactions between C₁₀H₁₅O₁₀ and C₁₀H₁₅O₄. This conclusion is made on the basis of RO₂ + RO₂ → ROOR + O₂ chemistry and the presence of C₁₀H₁₆O₉NO₃⁻ and C₁₀H₁₅O₉⁻ product ions in the MS/MS spectrum of C₂₀H₃₀O₁₂⁻ (Figure S12). However, examining this spectrum one can note another even more abundant product ion, i.e., C₁₀H₁₅O₆⁻, which leads me to conclude that the O₆ peroxy radical could also participate in the formation of the C₂₀H₃₀O₁₂ dimers. It is likely that the C₂₀H₃₀O₁₂ dimers are a mixture of products, where not only O₄ but also O₆ peroxy radicals play a key role in dimer formation.

-The detection of the C₁₀H₁₅O₆⁻ might suggest that the reaction of C₁₀H₁₅O₇ + C₁₀H₁₅O₇ contribute to the formation of C₂₀H₃₀O₁₂. However, this is less likely given that the C₁₀H₁₅O₇ RO₂ radical was not observed in the spectrum from α -pinene ozonolysis. As shown in the Figure R1, RO₂ radicals with odd number of oxygen atoms are much less abundant than RO₂ radicals with even number of oxygen atoms. Formation of C₂₀H₃₀O₁₂ might also arise from C₁₀H₁₅O₆ + C₁₀H₁₅O₈, as suggested by the reviewer, but the absence of product ions e.g., C₁₀H₁₄O₇ NO₃⁻ as observed for limonene might indicate that this reaction is minor. Finally, presence of the C₁₀H₁₅O₆⁻ product ion might arise from the fragmentation of a larger MS/MS product ion. Techniques having MSⁿ capabilities would help answering this question. Based on the available data, we cannot exclude that O₆ peroxy radicals are also involved in the formation of C₂₀H₃₀O₁₂.

Figure R1. Distribution of RO₂ radicals relative to the C₁₀H₁₅O₈ RO₂ radical observed during α -pinene and limonene ozonolysis. Data for RO₂ radicals with even oxygen atom numbers have already been shown in Figure S1c. Under our experimental conditions, RO₂ radicals with odd oxygen atom numbers are scarce as compared to the RO₂ radicals with even oxygen atom numbers. The abundance of C₁₀H₁₅O₇ and C₁₀H₁₅O₁₁ was negligible for the α -pinene system. The abundance of C₁₀H₁₅O₇ and C₁₀H₁₅O₉ was negligible for the limonene system.

-We have added discussions regarding the C₁₀H₁₅O₆⁻ ion and the possibility of additional dimer formation pathways. This section now reads as follows:

Page 14, lines 240-252: “For example, the reaction between C₁₀H₁₅O₁₀ and C₁₀H₁₅O₄ RO₂ radicals clearly contribute to the formation of C₂₀H₃₀O₁₂ dimer from α -pinene ozonolysis, according to the presence of C₁₀H₁₆O₉NO₃⁻ (m/z 342.0676) and C₁₀H₁₅O₉⁻ (m/z 279.0722) product ions in the MS/MS spectrum, as shown in Supplementary Figure S12. The C₁₀H₁₅O₆⁻ (m/z 231.0874) product ion observed in the MS/MS spectrum might suggest that RO₂ radicals other than O₄ and O₁₀ may contribute to the formation of dimer, although this product ion may have also originated from further fragmentation of a larger product ion. Hence, reactions between C₁₀H₁₅O₆ and C₁₀H₁₅O₈ radicals may also contribute to the formation of C₂₀H₃₀O₁₂ dimers, though the corresponding RO-OR product ions, e.g., C₁₀H₁₆O₅NO₃⁻ and C₁₀H₁₆O₇NO₃⁻, were not observed under our experimental conditions. The greater importance of the O₄ α -pinene RO₂ radical compared to the O₄ limonene RO₂ radical is also reflected in the dimer concentration distributions (Figure S1d), with less oxygenated dimers being more abundant during α -pinene ozonolysis (C₂₀H₃₀O₁₀ and C₂₀H₃₀O₁₂) than during limonene ozonolysis (C₂₀H₃₀O₁₂ and C₂₀H₃₀O₁₄).”

Line 198: It is not clear what is meant by a quaternary carbon bonded to two methyl groups. Perhaps it is best to refer to a scheme.

-We have highlighted in blue an example of this structure in Figure S9. The sentence now reads,

Page 12, lines 203-207: “In addition, the MS/MS spectrum of α -pinene C₁₀H₁₅O₈⁻ RO₂ radical displays two distinct fragmentation losses, CH₃O₂ and C₃H₆O, which are indicative of the methyl group specific to α -pinene RO₂, i.e., quaternary carbon bonded to two methyl groups (Figure S9, highlighted in blue), which is not present in the case of limonene RO₂.”

Technical comments (mainly related to MS terminology):

Lines 91-92 and several places elsewhere: I recommend to follow the IUPAC guidelines for terms related to mass spectrometry. The term “fragment ion” is deprecated and not the recommended term (only to be used in the case of fragmentation in the electron ionization mode); the recommended term

is “product ion”. The IUPAC guidelines can be found in the following paper: Ref.: K. K. Murray, R. K. Boyd, M. N. Eberlin, G. J. Langley, L. Li, Y. Naito. Definitions of terms relating to mass spectrometry. IUPAC Recommendations 2013. Pure Appl. Chem., 85, 1515-1609, 2013.

Line 94-95: same comment as above: replace “charged fragment” by “product ions”.

Line 101: MS/MS product ions

Line 116: observed product ions with relative abundance (Note: ions are abundant but ion signals are intense!).

Line 173: the product ion

Line 234: product ions in the MS/MS

Line 313: Product ion abundances were normalized to the maximum ion abundance

Line 315: 0.1% relative abundance

Lines 317-318: product ion relative abundances

Line 319: NO₃⁻ product ions, ...

Line 320: The precursor ion is the most abundant ion

Line 322: observed product ions

-We have replaced “fragment ion(s)” with “product ion(s)” and referred to the ion signal with “intensity” where applicable in texts and figures throughout the manuscript.

Line 363: Kerminen, V.-M. et al.

References: journal abbreviations should be checked; titles of journal articles should not be capitalized.

-We have corrected the abovementioned citation and updated the article titles in the reference section as suggested.

The supplement should also be checked for correct MS terminology.

-We have updated the MS terminologies and references in the Supplementary Information.

Reviewer #2 (Remarks to the Author):

The submitted work describes the results of CI-MS studies of oxidation products from selected natural hydrocarbons. The topic is important for the understanding of atmospheric particle formation and has been the subject of comparatively intensive research in recent years. The authors focus on two monoterpenes, which both play a special role in atmospheric particle formation. The experiments have been performed at a high level and provide interesting, undoubtedly publishable results about possible structures, reactivities and especially fragmentation paths of the observed ions. The manuscript is well written, the main references are cited and justified in relation to the conclusions drawn. My only point of criticism is to what extent the authors meet their own claim, namely "... (to) present the first experimental elucidation of RO₂ and dimer structures...". What the authors do show, however, are potential structures and suggestions of formation pathways that represent only the 'most plausible' structures of the fragments. In this respect the question arises if the manuscript represents an advance in understanding likely to influence thinking in the field or if it is simply a very good publication for e.g. a mass spectrometric journal.

-We thank the reviewer for the careful consideration of our study. Based on the reviewer comment, we have toned down the text in the Abstract and Introduction sections as detailed below. Although the current work does not provide a direct measure of the RO₂ and dimers structures, the combination of the fragmentation patterns with current knowledge of HOMs formation allow us to identify the preferential oxidation pathways for kinetic modeling of radical chemistry. Considering its broad impact and appeal beyond mass spectrometric developments, we believe that the current study aligns well with the scope of Nature Communications.

Change made to the abstract:

Page 3, Lines 34-35: "Here, we present the first experimental elucidation of RO₂ and dimer structures in the gas-phase, using online tandem mass spectrometry analyses"

is changed to

Page 3, Lines 33-35: "Here, we present the first in-situ characterization of RO₂ and dimer structure in the gas-phase using online tandem mass spectrometry analyses."

Changes made to the main text:

Page 5, Line 74-77 used to read "Here, we report the first online and in-situ structural elucidation of gas-phase HOMs, RO₂ radicals, and dimeric products. To access this unique information, we use our recently developed new analytical technique consisting of a NO₃⁻ chemical ionization (CI) source coupled to an ultrahigh-resolution Orbitrap mass spectrometer.^{38,39,}"

This is now changed to

Page 5, Line 74-77: "Here, we report the first in-situ structural characterization of gas-phase HOMs, RO₂ radicals, and dimeric products using a newly developed analytical technique based on an NO₃⁻ chemical ionization (CI) source coupled to an ultrahigh-resolution Orbitrap mass spectrometer.^{39,40} Using MS/MS data, we were able to infer the most plausible isomers of different compounds."

Page 5, Line 80-83 used to read: "In this work, this technique is applied to characterize the structure of closed-shell products and radicals from monoterpene oxidation. Based on the structural information gained by the technique, we constrain the main chemical routes involved in autoxidation and dimer formation."

This is now changed to

Page 5, Line 90-92: “We propose the most plausible oxidation product structures, autoxidation mechanism, and dimer formation pathways based on the observed MS/MS product ions and neutral losses for the two monoterpene precursors.”

-Page 16, Lines 281-283 used to read “When applied to gaseous dimers, online MS/MS reveals unique RO-OR dimer fragmentation pathways that help to elucidate the identity of the dominant RO₂ radical precursors, which is further supported by hierarchical clustering analyses.”

This is now changed to

-Page 16, Lines 281-283 “When applied to gaseous dimers, online MS/MS reveals unique RO-OR dimer fragmentation pathways that help identifying main RO₂ radical precursors, which is further supported by hierarchical clustering analyses.”

Reviewer #3 (Remarks to the Author):

Dear Editor, dear authors,

I thoroughly read the article Structures and Reactivity of Peroxy Radicals and Dimeric Products Revealed by Online Tandem Mass Spectrometry by Sophie Tomaz , Dongyu Wang et al submitted to Nature Communications (NCOMMS-20-29025-T).

The study presents new and certainly relevant data on the structure analysis of highly oxygenated organic molecules (HOMs), which contribute to the formation and growth of atmospheric particles. Specifically, the authors report new results on the mechanism and reaction pathways of organic peroxy radicals (RO₂) which are extremely reactive compounds, that are of vital importance for the atmospheric degradation of volatile hydrocarbons. Ultimately, highly oxygenated organic molecules (HOMs) are formed, including low-volatility ROOR dimers generated by bimolecular RO₂ + RO₂ reactions. In the report, a promising experimental approach by online tandem mass spectrometry is presented to investigate the ions of interest in the gas-phase. The authors claim that their analytical strategy allows to constrain the structures and formation pathway of several HOM-RO₂ radicals and dimers produced from monoterpene ozonolysis, a prominent atmospheric oxidation process.

At this point it is important to note that the authors report extended mechanisms on the basis of their CID experiments of the NO₃⁻ adduct molecular ions but the majority of the structures shown are: tentative or most probable isomers – which is ok and correct as the ion structures are not proven besides their composition meeting the accurate ion mass ... This problem is certainly a known one for this kind of studies and does not diminish the value of the work, which is in my eyes still respectable and well conducted. However, I strongly recommend to tone down the wording in the abstract and the intro part.

There is no final and exhaustive evidence for the ion structures found in the paper. The recently introduced analytical technique comprises a NO₃⁻ - chemical ionization (CI) source coupled to an ultrahigh-resolution Orbitrap mass spectrometer. In addition to its high mass resolution (140 000 at m/z 200 for MS and 70 000 at m/z 200 for MS/MS analysis at FWHM) and fast time resolution (10s), the CI-Orbitrap is also capable of performing online MS/MS analysis of compounds at sub-parts per trillion by volume (pptv) level. This powerful set-up was already introduced and reported in two previous papers by the laboratory (see refs 38 and 39).

-We thank the reviewer for the careful consideration of our study. Based on the reviewer's comment we have toned down the abstract, introduction and discussion sections as explained in the following:

Page 3, Lines 34-35: "Here, we present the first experimental elucidation of RO₂ and dimer structures in the gas-phase, using online tandem mass spectrometry analyses"

is changed to

Page 3, Lines 33-35: "Here, we present the first in-situ characterization of RO₂ and dimer structure in the gas-phase using online tandem mass spectrometry analyses."

-Page 5, Lines 74-77: "Here, we report the first online and in-situ structural elucidation of gas-phase HOMs, RO₂ radicals, and dimeric products. To access this unique information, we use our recently developed new analytical technique consisting of a NO₃⁻ chemical ionization (CI) source coupled to an ultrahigh-resolution Orbitrap mass spectrometer.^{38,39}"

is changed to

Page 5, Lines 74-77: “Here, we report the first in-situ structural characterization of gas-phase HOMs, RO₂ radicals, and dimeric products using a newly developed analytical technique based on an NO₃⁻ chemical ionization (CI) source coupled to an ultrahigh-resolution Orbitrap mass spectrometer.^{39,40}”

-Page 5, Line 80-81: “In this work, this technique is applied to characterize the structure of closed-shell products and radicals from monoterpene oxidation. Based on the structural information gained by the technique, we constrain the main chemical routes involved in autoxidation and dimer formation”

This has been changed, including an explanation for the selection of α -pinene and limonene as suggested by Reviewer #1.

Page 5, Lines 81-92: “In this work, this technique is applied to characterize the structure of closed-shell products and radicals produced from the oxidation of two monoterpenes, α -pinene and limonene. α -Pinene was chosen due to its global emission reaching up to 66.1 Tg yr⁻¹ and its importance in the formation of organic aerosol.⁴¹ Despite the lower emission rate of limonene (11.4 Tg yr⁻¹), limonene has a greater reactivity towards ozone and a higher HOM yield than α -pinene.¹⁰ As a result, limonene ozonolysis is expected to play a major role in new particle formation. In addition, limonene is found in cleaning and personal care products, and might be a significant source for indoor secondary organic aerosols (SOA).⁴¹⁻⁴⁴ Finally, limonene has a single cyclic structure as opposed to the bicyclic structure of α -pinene, which simplifies the interpretation of the MS/MS fragmentation patterns. We propose the most plausible oxidation product structures, autoxidation mechanism, and dimer formation pathways based on the observed MS/MS product ions and neutral losses for the two monoterpene precursors.”

-Page 16, Lines 281-283 “When applied to gaseous dimers, online MS/MS reveals unique RO-OR dimer fragmentation pathways that help to elucidate the identity of the dominant RO₂ radical precursors, which is further supported by hierarchical clustering analyses.”

Is changed to

-Page 16, Lines 281-283 “When applied to gaseous dimers, online MS/MS reveals unique RO-OR dimer fragmentation pathways that help identifying main RO₂ radical precursors, which is further supported by hierarchical clustering analyses.”

-We have also added instrumental details in the Methods and Supplementary information sections, including clarification of the MS/MS method employed. We would also like to point out that the MS/MS experiments performed in this work were conducted with a higher energy collision dissociation (HCD) cell after mass isolations of the parent ion (± 0.4 Da) using the quadrupole, which is different from in-source CID where the parent ion identities are more uncertain.

Page 18, Lines 321-329: “Higher energy collision dissociation (HCD) was used to obtain targeted MS/MS spectra using a quadrupole ion isolation window width of 0.4 Da. No in-source collision-induced dissociation (CID) energy was applied. The HCD cell is located after the quadrupole capable of mass isolation within ± 0.4 Da. The selectivity of the NO₃⁻ detection scheme towards acidic and highly oxidized compounds ensures that a single parent ion is selected for structural characterization using MS/MS acquisitions for the RO₂ radicals and dimeric products discussed in the main text within the mass isolation window. In the case that more than one parent ion is isolated, the dominant parent ion is at least an order of magnitude more abundant than the other parent ion(s) in most cases.”

Page 18, Lines 336-341 “Peak identification and assignment were performed using XCalibur 4.1 (Thermo Scientific). Identification of parent ion formula was constrained based on known oxidation chemistry and elemental composition, i.e., ions are assumed to contain only carbon, hydrogen, and oxygen atoms with up to 2 nitrogen atoms to account for adduction with NO₃⁻ or HNO₃NO₃⁻. Identification of MS/MS product ion is constrained by the parent ion elemental composition.”

-We have also added information about the mass accuracy of the instrument. Please also see our response to the comment below for additional details.

Page 5, Lines 77-80: “In addition to its high mass resolution (140 000 at m/z 200 for MS and 70 000 at m/z 200 for MS/MS analysis at FWHM) and fast time resolution (10s), the CI-Orbitrap is also capable of performing online MS/MS analysis of compounds at sub-parts per trillion by volume (pptv) level.”

is changed to

Page 5, Lines 78-81: “The CI-Orbitrap is capable of performing online MS/MS analysis of compounds at sub-parts per trillion by volume level with high mass resolution (70,000 at m/z 200 at FWHM) and high mass accuracy (≤ 5 parts per million).”

In this regard I strongly recommend to report only accurate ion masses with 5 decimal places (!) when these values are justified by minimal experimental errors of the respective accurate ion mass measurement. This mandatory information is missing. Without this information these values are not acceptable. High resolution capability alone is not the key information needed here – It is important to also report how accurate and precise your measurements are.

-As described in additional methods, the Orbitrap used for online MS/MS in negative ion mode was calibrated using an aqueous solution of sodium acetate (2mM, Aldrich, > 99% purity). During online CI-measurements, the drift in mass accuracy is negligible. For instance, the mass accuracy from MS/MS spectra of parent ions over the course of ~12 hours of continuous measurement was within 2 ppm, as shown below.

Figure R2. Mass accuracy for parent ions (solid circles) and CI reagent ion (NO_3^- , hollow square) observed over the course of a ~12 hours continuous measurement period. The mass error was calculated as the difference between the theoretical m/z value and the experimental m/z value from the 30-minute-average MS/MS spectra for each of the parent ions. NO_3^- is observed in all MS/MS spectra, and the error bar represents the standard deviation of its mass error (~ 0.2 ppm).

As described in Line 342-343, ions with low abundance and/or m/z error exceeding 5 ppm are not reported or included in the interpretation of MS/MS. Following the comment from the reviewer, we have decided to reduce the number of decimals to four comprising the potential experimental errors. Also, constraints were placed on the molecular formula search for parent ions based on known oxidation chemistry. We have added the information about m/z calibration and mass accuracy to the methods section in the main text,

Page 18, Lines 332-336: “The Orbitrap was mass-calibrated using an aqueous sodium acetate solution (2mM, Aldrich, > 99% purity). During online CI-measurements, the drift in mass accuracy was negligible (i.e., < 2ppm) based on the reported m/z of the chemical ionization reagent ions (NO_3^-

and $\text{HNO}_3\text{NO}_3^-$) and m/z values from MS/MS parent ions, which is within the specifications of the Orbitrap.”

In conclusion I strongly recommend to clarify the wording on the ion structure assumptions in the discussions of the mechanisms. Additionally, the lacking infos needed to complete the experimental part on the accurate ion mass determinations have to be given before a final decision on the publication can be drawn.

-As discussed in the previous comments, we have toned down the abstract and the introduction parts. We have also added details about the calibration and mass accuracy to the Methods section in the text, and updated the supplementary information section accordingly.

REVIEWERS' COMMENTS

Reviewer #1 (Remarks to the Author):

The authors have properly addressed my concerns and I have no further comments.
Magda Claeys

Reviewer #3 (Remarks to the Author):

Dear Authors,
the revision satisfies my critique.

The paper is now ready to go.

Whether it is suited for publication in nature communications is up to the editor - I could also imagine to see this paper published in a MS Journal as reviewer II suggests.

Responses to Reviewer Comments

We thank the reviewers for their comments and suggestions, reprinted below in blue. As both reviewers have accepted the paper without additional comments/edits, no action required.

Reviewer #1 (Remarks to the Author):

The authors have properly addressed my concerns and I have no further comments.
Magda Claeys

Reviewer #3 (Remarks to the Author):

Dear Authors,
the revision satisfies my critique.

The paper is now ready to go.

Whether it is suited for publication in nature communications is up to the editor - I could also imagine to see this paper published in a MS Journal as reviewer II suggests.